# Experimental *Schistosoma japonicum*-induced pulmonary hypertension

**Biruk Kassa**[1] *, **Michael H. Lee**[1], **Rahul Kumar**[1], **Claudia Mickael**[2], **Linda Sanders**[2], **Rubin M. Tuder**[2], **Margaret Mentink-Kane**[3], **Brian B. Graham**[1] *

**1** Department of Medicine, University of California San Francisco, San Francisco, California, United States of America, **2** Department of Medicine, University of Colorado Anschutz Medical Campus, Aurora, Colorado, United States of America, **3** Biomedical Research Institute, Rockville, Maryland, United States of America

* biruk.kassa@ucsf.edu (BK); brian.graham@ucsf.edu (BBG)

**Data Availability Statement:** All relevant data are within the manuscript and its Supporting Information files.

## Abstract

### Background

Schistosomiasis, a major cause of pulmonary arterial hypertension (PAH) worldwide, is most clearly described complicating infection by one species, *Schistosoma mansoni*. Controlled exposure of mice can be used to induce Type 2 inflammation-dependent *S. mansoni* pulmonary hypertension (PH). We sought to determine if another common species, *S. japonicum*, can also cause experimental PH.

### Methods

Schistosome eggs were obtained from infected mice, and administered by intraperitoneal sensitization followed by intravenous challenge to experimental mice, which underwent right heart catheterization and tissue analysis.

### Results

*S. japonicum* sensitized and challenged mice developed PH, which was milder than that following *S. mansoni* sensitization and challenge. The degree of pulmonary vascular remodeling and Type 2 inflammation in the lungs was similarly proportionate. Cross-sensitization revealed that antigens from either species are sufficient to sensitize for intravenous challenge with either egg, and the degree of PH severity depended on primarily the species used for intravenous challenge. Compared to a relatively uniform distribution of *S. mansoni* eggs, *S. japonicum* eggs were observed in clusters in the lungs.

### Conclusions

*S. japonicum* can induce experimental PH, which is milder than that resulting from comparable *S. mansoni* exposure. This difference may result from the distribution of eggs in the lungs, and is independent of which species is used for sensitization. This result is consistent with the clearer association between *S. mansoni* infection and the development of schistosomiasis-associated PAH in humans.

**Funding:** Grant funding was provided by the American Heart Association Grant 19CDA34730030 (RK), ATS Foundation/Pulmonary Hypertension Association Research Fellowship (RK) and The Cardiovascular Medical Research Fund (CMREF; RK); NIH Grant F32HL151076 to MHL; 2020 Entelligence Award (Actelion Pharmaceuticals) and PRIDE-AGOLD (NIH R25HL14166) (CM); and NIH Grants P01HL152961 (RMT and BBG) and R01HL135872 (BBG). The funders had no role in study design, data collection and analysis, decision to publish, or preparation of the manuscript.

**Competing interests:** The authors have declared that no competing interests exist.

## Author summary

One fatal complication of chronic schistosomiasis is the development of pulmonary hypertension, a disease of progressive narrowing and obstruction of the lung blood vessels leading to heart failure. How schistosomiasis causes pulmonary hypertension is not well understood. Infection with *Schistosoma mansoni* is mostly clearly associated with pulmonary hypertension, and it is unclear if other species can also cause pulmonary hypertension. In this study, we experimentally exposed mice to *Schistosoma japonicum*, and found that they can develop pulmonary hypertension, but the severity is less than in mice exposed to *Schistosoma mansoni*. This may help explain why there are only a few human cases described of pulmonary hypertension following *Schistosoma japonicum*. We also investigated similarities and differences between the antigenic triggers of inflammation, and found that *Schistosoma mansoni* and *japonicum* antigens were substitutable for each other.

## Introduction

Schistosomiasis is a neglected tropical disease of high global prevalence, caused by blood flukes in the genus *Schistosoma* [1]. Although easily curable at the acute infection stage, subsequent maturation of the *Schistosoma* cercariae in the portal venous system and their egg production can cause chronic long-term organ injury in the human host, including hepatosplenic schistosomiasis (SchHSD) and schistosomiasis-induced pulmonary arterial hypertension (SchPAH) [2,3]. Due to the high prevalence of schistosomiasis, SchPAH is a major cause of WHO Group 1 pulmonary arterial hypertension (PAH) globally [4].

Similar to other PAH etiologies, SchPAH is characterized by elevated pulmonary vascular resistance; characteristic pulmonary vascular histopathology including plexiform lesions; right ventricular dysfunction; and clinical response to pulmonary vasodilator therapy [1]. SchHSD causes portal hypertension and egg embolization to the lungs via portocaval shunts [5], and is likely a major risk factor for the development of SchPAH. While several mechanisms could contribute to SchPAH pathogenesis in humans, including pulmonary vascular obstruction by the *Schistosoma* eggs and portopulmonary hypertension, SchPAH is thought be a predominantly inflammatory vasculopathy [6]. Our group previously demonstrated that in a mouse model, Th2 inflammation is the key pathogenetic mechanism by which *Schistosoma mansoni* eggs cause experimental pulmonary hypertension (PH), as deletion of both IL-4 and IL-13 suppresses the *Schistosoma*-PH phenotype in mice [7]. In this model, pulmonary vascular inflammation is triggered by initial intraperitoneal sensitization followed by intravenous challenge with *S. mansoni* eggs [8], mimicking the peri-portal egg embolization occurring in chronic infection. Increased pulmonary Th2 inflammation is also observed in humans with *S. mansoni*-induced PAH [7,9]. The Type 2 inflammation mechanistically triggers recruitment of thrombospondin-1-expressing monocytes to the pulmonary adventitia, causing localized activation of pathologic TGF-β [10,11], a signaling pathway shared with other PAH etiologies.

Although our understanding of SchPAH pathogenesis continues to expand, important knowledge gaps still exist. One of these is which *Schistosoma* species can cause PAH, and how disease caused by different species relates to one another. Three *Schistosoma* species cause ~95% of clinical schistosomiasis: *S. mansoni*, *S. haematobium*, and *S. japonicum* (**Fig 1A**). The worms of *S. mansoni* and *S. japonicum* migrate to the portal venous system, where they mate

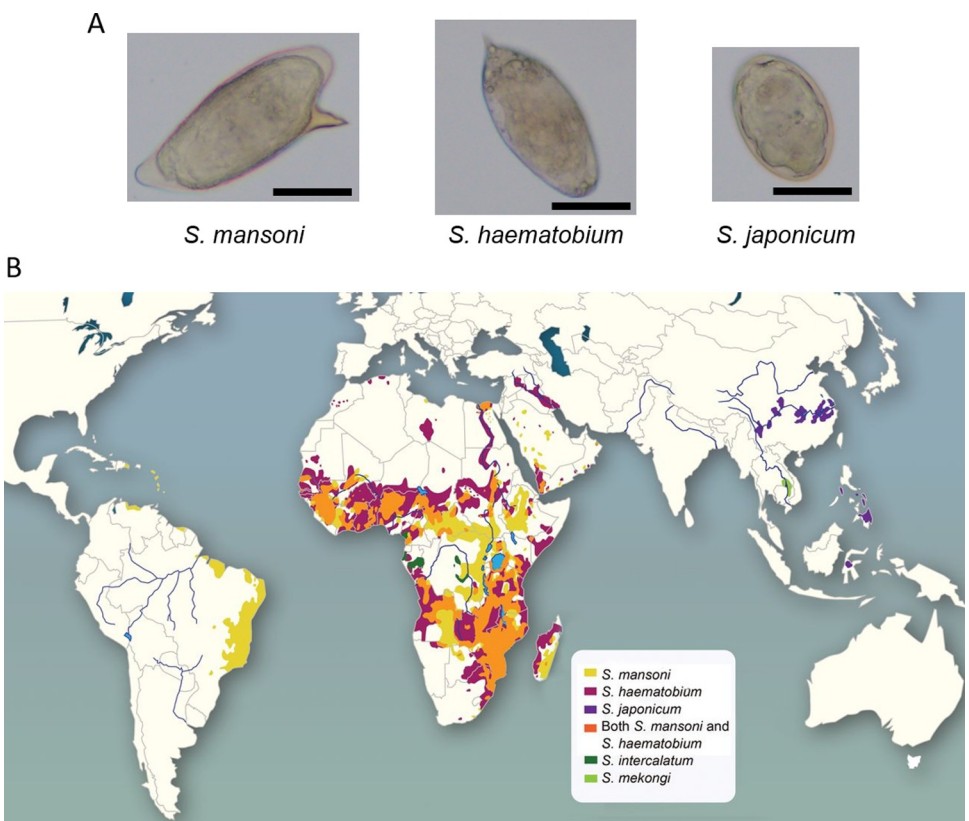

**Fig 1. *S. mansoni*, *S. haematobium*, and *S. japonicum* are the three most common Schistosoma species that infect humans.** (A) Pictures of eggs of each species (all scale bars = 50μm). The species can be distinguished by the egg shape (oval or round) and location of the spike (angled, or at one end). (B) The geographical distribution of *Schistosoma* species (reproduced with permission from Weerakoon, et al. [12]).

and lay eggs, resulting in hepatointestinal disease. The species have distinct geographic distributions, mediated by where the intermediate host snail lives, with *S. mansoni* endemic in Brazil and several other south American countries, and throughout sub-Saharan Africa; and *S. japonium* in east Asia including China, the Philippines, and Indonesia [12] (**Fig 1B**). *S. mansoni* is the species most clearly associated with PAH [13], and other species much less so, but it is unclear if there are biases in testing or reporting based on geographic locations, or a true pathobiological basis for species-dependent differences in PAH prevalence resulting from biologic phenomena. For example, decreased prevalence of SchPAH with *S. japonium* infection could represent weaker immunogenicity of *S. japonicum* egg antigens—which would in turn shed light on specific antigen(s) and mechanisms which drive SchPAH. This uncertainty also has substantial public health implications, such as identifying at-risk populations to screen for PAH.

We therefore sought to determine if experimental PH in mice can result from controlled exposure to *S. japonicum* eggs, and how hemodynamic and inflammatory endpoints compare to *S. mansoni*-induced experimental PH. We also sought to determine if there may be antigenic similarity between the eggs of the two species, as a step towards characterizing and identifying specific antigens that drive inflammatory pulmonary vascular disease.

## Methods

### Ethics statement

All animal studies were approved by the University of California San Francisco Institutional Animal Care and Use Committee, Protocol AN181431. The procedures followed were in accordance with institutional guidelines.

### Animal models

C57BL6/J background wild-type mice were purchased from Jackson Laboratories (Bar Harbor, ME). The mice used were female, and between 6 and 8 weeks of age at the start of the experiment. All animals were housed under specific pathogen-free conditions in an American Association for the Accreditation of Laboratory Animal Care-approved facility of University of California San Francisco.

### Model of *Schistosoma* egg-induced PH

Fresh *Schistosoma mansoni* (NMRI strain) and *japonicum* (Chinese strain) eggs were harvested from the liver of previously infected Swiss Webster mice provided by Biomedical Research Institute (BRI, Rockville MD) using standard techniques [14]. Mice were intraperitoneally (IP) sensitized with 240 *Schistosoma* eggs per gram body weight followed by intravenous (IV) challenge with 175 *Schistosoma* eggs per gram body weight 14 days later, as we have previously done for *S. mansoni* eggs [7,8,11,15]. Other groups were challenged without being previously sensitized.

### Right ventricular systolic pressure (RVSP) and right ventricular hypertrophy assessments

Right heart catheterization was conducted to measure PH and RV hypertrophy using standard techniques [8,11]. In brief, mice were anesthetized, tracheostomy and mechanical ventilation performed, and the abdomen and diaphragm opened. A 1 Fr pressure-volume catheter (Millar, Houston, TX) was inserted into the RV by direct puncture through the RV free wall to measure the RVSP, and then into the LV. Then, the lungs were flushed with PBS, and the right lung hilum sutured followed by instillation with 1% low melt agarose to inflate the left lung prior to embedding in paraffin for histology. The right lung was snap frozen for protein quantification.

### Vascular remodeling quantification

Formalin fixed and paraffin embedded (FFPE) tissue was immunofluorescence stained for α-smooth muscle actin-stained vascular vessels using the protocol outlined in **Table 1**. Images were captured using a Nikon Eclipse 80i microscope (Nikon, Melville, NY) and Olympus DP74 color camera (Olympus, Waltham, MA). The vascular media was identified by thresholding and tracing using image processing software (Image-Pro 10, Media Cybernetics,

**Table 1. Immunostaining reagents and protocol.**

| Immuno-stain | Antigen Retrieval | Block | Primary Antibody | Secondary Antibody | Tertiary Reagent | Mounting Reagent |
|---|---|---|---|---|---|---|
| Anti-Alpha-Smooth Muscle Actin (α-SMA) (Invitrogen 14-9760-82) | Borg Decloaker RTU, 20min boiling (BioCARE MEDICAL BD1000G1); TBST rinses | M.O.M mouse IgG blocking reagent (Vector BMK-2202) 2 drops in 2.5 ml TBS, 1hr at RT | 1:200 in M.O.M diluent (1:13.3 protein concentrate in TBS, Vector BMK-2202), 1hr at RT | M.O.M Biotinylated anti-mouse IgG (Vector BMK-2202) in M.O.M diluent, 10min at RT | 1:500 Streptavidin Fluorescein Conjugate (Invitrogen S-869) in TBS, 30min at RT | Vectashield with DAPI (Vector H-1500) |

Rockville, MD), the effective radii of the outer and internal perimeters of the medial layer quantified, and the fractional media thickness calculated as the difference in radii divided by the external media radius.

### Granuloma analysis

The optical rotator sterological method was used to estimate the peri-egg granuloma volume [16] on images of granulomas from hematoxylin and eosin (H&E) stained slides surrounding a single egg were captured and analyzed using image processing software (Image-Pro 10), using the egg as the center reference point. The total number of granulomas visualized in lung sections was counted (each of which may contain no eggs or any number of eggs), per $mm^2$ of lung tissue analyzed.

### Protein assessment and ELISA

Whole-lung tissue lysates were extracted from snap-frozen lung tissue macerated and sonicated in RIPA buffer containing anti-proteases, as previously performed [11]. The obtained lysate was used to quantify IL-4 and IL-13 protein concentrations in enzyme-linked immunosorbent assay (ELISA) using standard approaches (kits M4000B and M1300CB, R&D Systems, Minneapolis, MN). Protein concentrations were normalized by total protein (Bradford assay, Bio-Rad, Hercules, CA)

### Egg counts

To quantify the residual egg burden in the lungs of experimental mice, 15–20 mg of lung tissue was dissolved using 4% KOH per gram, incubated for 18 hours in a 37°C shaker incubator to digest the tissue without damaging the eggs [17]. The eggs were then counted using a Nikon Labophot-2 microscope.

### Statistics

Statistical software (GraphPad Prism v9, San Diego, CA) was used for statistical analysis and graph generation. Statistical difference between two groups was assessed by t-test, and one-way ANOVA was used to assess differences between more than 2 groups, followed by post-hoc Tukey test. *P* values <0.05 were considered statistically significant.

## Results

### *S. japonicum* can cause experimental PH, which is less severe than *S. mansoni*-induced PH

To determine if *S. japonicum* could cause experimental PH, we sensitized and challenged mice with *S. japonicum* eggs (as outlined in **Fig 2A**). PH is foremost characterized by elevated pressures in the right heart and pulmonary arteries. We observed that mice sensitized and challenged with *S. japonicum* developed PH as evidenced by an increase in right ventricle systolic pressure (RVSP) (**Fig 2B**). Elevated pressures may result from a combination of vascular remodeling and vasoconstriction [10]. We observed increased media thickness in the pulmonary arteries in mice with *S. japonicum*-induced PH (**Fig 2C**). The media thickness was on average comparable and had a similar distribution in the *S. japonicum* and *S. mansoni* groups.

We compared *S. japonicum* sensitized and challenged mice to those which had been sensitized and challenged with *S. mansoni* eggs. We observed that *S. japonicum* induced milder PH as assessed by RVSP than the PH induced by *S. mansoni* (**Fig 2B**).

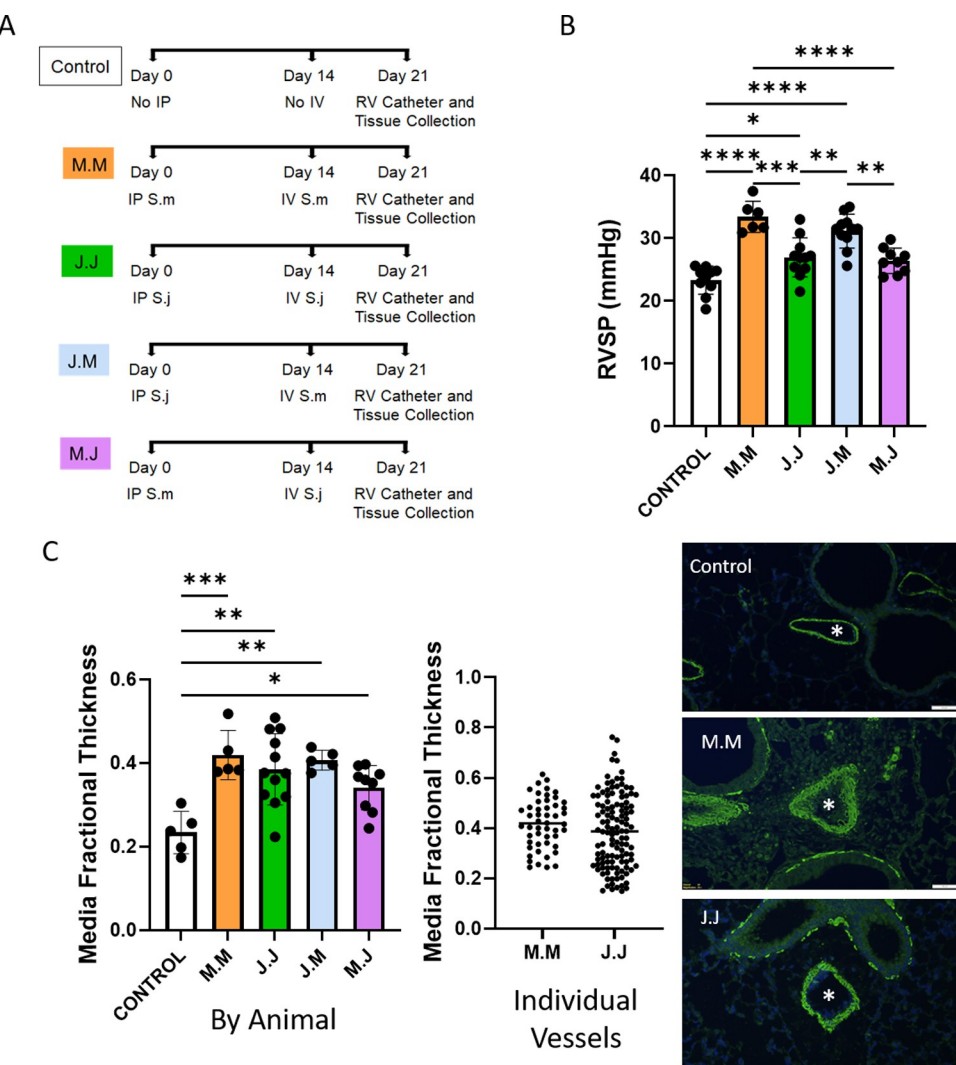

**Fig 2.** *Schistosoma japonicum* **can cause experimental PH in mice, which is milder than** *Schistosoma mansoni*-**induced PH.** (A) Experimental outline for different groups used to test the role of different *Schistosoma* species in inducing experimental PH, indicating intraperitoneal (IP) sensitization exposure, intravenous (IV) challenge exposure, and right ventricle (RV) catheterization and tissue collection. (B) Right ventricular systolic pressure (RVSP) of *Schistosoma* exposed mice, and control mice, indicative of pulmonary hypertension severity; n = 6–11 per group; ANOVA P<0.0001. (C) Vascular media thickness of *Schistosoma* exposed and control mice, indicative of vessel remodeling; n = 5–12 animals per group, and n = 50, 120 vessels; ANOVA P = 0.0003. Representative images are shown. *: vessel lumen. Scale bar: 50 μm. Graphs show mean ± SD; post-hoc Tukey tests shown; *P<0.05, **P<0.01, ***P<0.005, ****P<0.001.

## *S. japonicum* and *S. mansoni* antigens are cross-reactive in causing PH

We next sought to determine if antigens for the two species could be cross-reactive, that is, if sensitization by one parasite can induce the immunologically-driven PH phenotype after intravenous challenge with the other parasite (**Fig 2A**). We found that mice that were sensitized with either species first, followed by challenge with the other species, all had an increased RVSP compared to control mice (**Fig 2B**). Notably, the degree of RVSP appeared to be dependent on only the species used for intravenous challenge—that is, the degree of PH was highest

when the mice were intravenously challenged with *S. mansoni*, independent of which species were used for sensitization.

## *S. japonicum* induces attenuated Type 2 inflammation compared to *S. mansoni*

We quantified the overall severity of peri-egg inflammation by estimating the volume of granulomas around single observed eggs using stereology. We observed the *S. japonicum* granulomas were 35-fold smaller on average than the *S. mansoni* granulomas (**Fig 3A**). We next quantified the degree of Type 2 inflammation by measuring the concentration of IL-4 and IL-13 protein in whole lung lysates by ELISA. We observed that, compared to control, there was increased IL-4 concentrations in the sensitized and challenged groups, with roughly similar levels between groups (**Fig 3B**). We observed the highest IL-13 concentrations in the mice that received intravenous *S. mansoni*, and were particularly low in the group sensitized with *S. mansoni* and challenged with *S. japonicum* (**Fig 3C**).

## *S. japonicum* eggs tend to cluster more than *S. mansoni*

One possible mechanism by which *S. japonicum* could induce weaker PH is more rapid clearance of the eggs by the host immune system, but we observed that the total residual egg burden in each group was similar (**Fig 4A**), as was the number of granulomas visualized in tissue sections (**Fig 4B**).

We did observe that the intravenously administered *S. japonicum* eggs appeared to cluster in the lung tissue, in contrast to a more uniform distribution of *S. mansoni* eggs (**Fig 4C**).

## Intravenous eggs from either species alone do not cause PH, and much less inflammation

As control groups, we performed intravenous challenge alone, with either species (**Fig 5A**). We observed that there was no significant increase in RVSP in mice after intravenous challenge alone (**Fig 5B**). We similarly found there was no significant vascular remodeling (**Fig 5C**).

We assessed the estimated peri-egg granuloma volumes in the mice that received only intravenous eggs, finding that mice challenged only with *S. mansoni* have approximately 3-fold larger peri-egg granuloma volumes than mice challenged only with *S. japonicum* (**Fig 6A**). The IV *S. mansoni* only granulomas were 39-fold smaller than those from mice previously sensitized with *S. mansoni*, and the IV *S. japonicum* only granulomas were 3.6-fold smaller than those from mice previously sensitized with *S. japonicum*. We similarly observed that intravenous eggs alone caused mild increases in IL-4 and IL-13, to levels not significantly higher compared to control mice (**Fig 6B and 6C**). The number of residual eggs 7 days after intravenous eggs alone (with either species) was similar to the numbers in mice that received prior sensitization (**Fig 6D**).

## Discussion

For the first time, to our knowledge, we demonstrated that inflammatory PH can occur in mice following experimental exposure to *S. japonicum* eggs. Our approach used intraperitoneal sensitization followed by intravenous challenge. Importantly, both sensitization and challenge were required: intravenous eggs alone were insufficient. This observation is consistent with the concept that *Schistosoma*-PH in mice is a primarily immunologically-driven phenomenon, as may also be true for *Schistosoma*-PAH in humans.

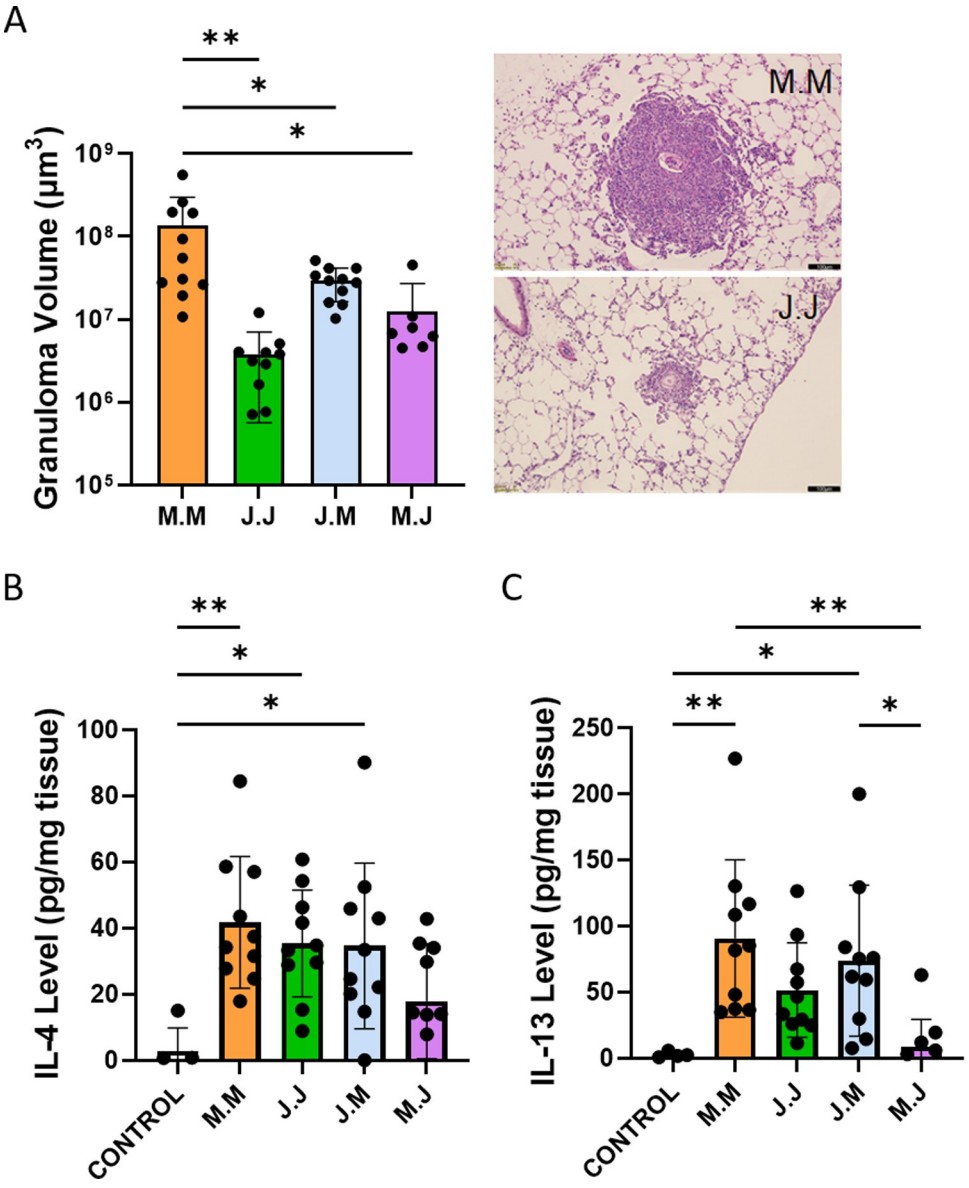

**Fig 3.** *Schistosoma japonicum* **induces mild Type 2 inflammation in mice.** (A) Estimated granuloma volumes of *Schistosoma* exposed mice, as a readout of overall inflammation; n = 7–11 per group; ANOVA P = 0.007. Representative images are shown. Scale bar: 100 μm. (B) IL-4 protein concentration in *Schistosoma* exposed and control mice; n = 5–10 per group; ANOVA P = 0.0028. (C) IL-13 protein concentration in *Schistosoma* exposed and control mice; n = 5–10 per group; ANOVA P = 0.0003. Graphs show mean ± SD; post-hoc Tukey tests shown; * P<0.05, ** P<0.01.

We observed that the severity of PH, as primarily assessed by RVSP, appeared to be less pronounced in the *S. japonicum* model compared to those in the *S. mansoni* model, following the same degree of egg sensitization and challenge. The degree of vascular remodeling was comparable, suggesting a difference in vasoconstriction, which we have previously found to be Rho-kinase mediated [10]. The RVSP difference suggests that there is a pathobiological difference between eggs of the two species, which could result in greater incidence and prevalence of PAH in humans infected with *S. mansoni* compared to those infected with *S. japonicum*. This possibility is supported by the limited epidemiologic literature available in this field. One study

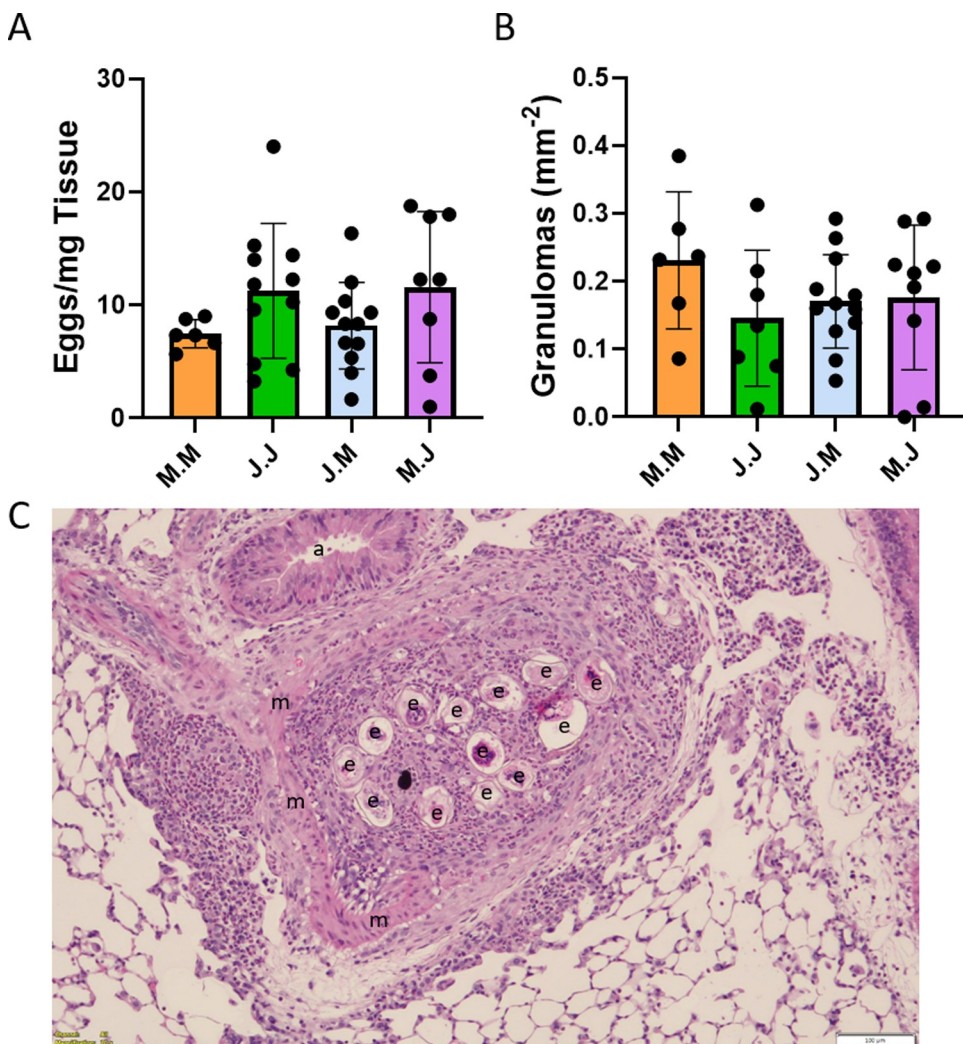

**Fig 4. *Schistosoma japonicum* tends to form clusters during infection.** (A) Residual *Schistosoma* egg burden in lung samples from *Schistosoma*-exposed mice at the time of right heart catheterization; n = 6–12 per group; ANOVA P = NS; graph shows mean ± SD. (B) Number of granulomas visualized per mm$^2$ of lung tissue analyzed (which is dependent on both egg count and granuloma size); n = 6–12 per group; ANOVA P = NS; graph shows mean ± SD. (C) Representative image showing *Schistosoma japonicum* egg clusters in the lungs, surrounded by significant inflammation within and around the vessel (scale bar = 100μm). Labels: e = *S. japonicum* eggs; m = medial layer of the vessel, a = airway.

in Recife, Brazil from 2009 (only *S. mansoni* is endemic in Brazil) using a threshold of 40mmHg on echocardiography found that 9 of 84 subjects with SchHSD had evidence of pulmonary hypertension, for a calculated prevalence of 11% [18]. Another study from São Paulo, Brazil from 2009 found 12 of 65 patients with SchHSD had a systolic pulmonary artery pressure >40mmHg on screening echocardiography (18%) [3]. In contrast, a recently reported retrospective series from Changsha in south-central China (where only *S. japonicum* is endemic) found only 10 individuals with a history of schistosomiasis who also had had echocardiograms with RVSP>40mmHg, out of 18,829 total individuals with schistosomiasis, or a prevalence of 0.053% [19]. In combination, our preclinical studies here and these reported clinical data together support the concept that *S. japonicum* likely causes SchPAH less often or to a milder degree than *S. mansoni*.

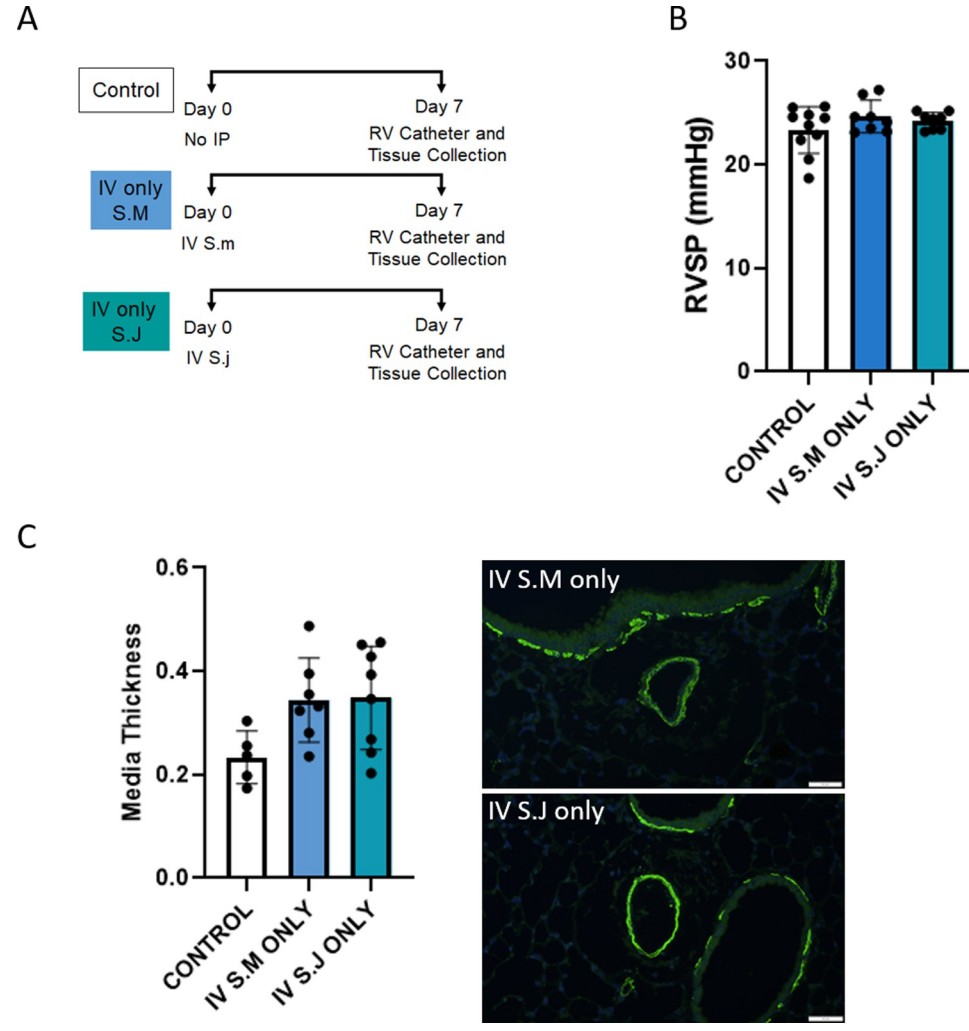

**Fig 5. Intravenous egg exposure alone from *Schistosoma* species do not cause PH.** The control group data presented in panels B and C of this figure are the same control data presented in Fig 1. (A) Experimental timeline for testing induction of PH with intravenous egg exposure alone. (B) Right ventricular systolic pressure (RVSP) of intravenous egg alone and control mice; n = 8–10 per group; ANOVA P = NS. (C) Vascular media thickness of intravenous egg alone and control mice; n = 5–8 per group; ANOVA P = NS. Graphs show mean ± SD. Representative images are shown. Scale bar: 50 μm.

Here we used the same egg dose in comparing *S. japonicum* and *mansoni*. Using the same number of eggs for the two species does not reflect the fact that *S. japonicum* has substantially higher fecundity than *S. mansoni*: *S. japonicum* worm pairs release ~5000 eggs/worm pair/day, versus ~200 eggs/worm pair/day for *S. mansoni* [20,21]. It is thus possible that humans chronically infected could have higher total egg burden in *S. japonicum* infection if they had comparable worm burden, and egg burden in the lungs is likely to correlate with PAH development.

While the precise mechanism for why *S. mansoni* eggs more potently trigger PH remains unexplored, biological differences between the two eggs provide a few plausible explanations. Eggs of the two species are of different shapes, with *S. mansoni* eggs being round, and *S. japonicum* eggs ovoid, but the short-axis diameters are similar, so the vessel diameters in which the eggs deposit are likely to be the same. It has been reported, as we observed, that *S. japonicum* eggs tend to cluster in tissues, in contrast to the solitary distribution of *S. mansoni* eggs [20].

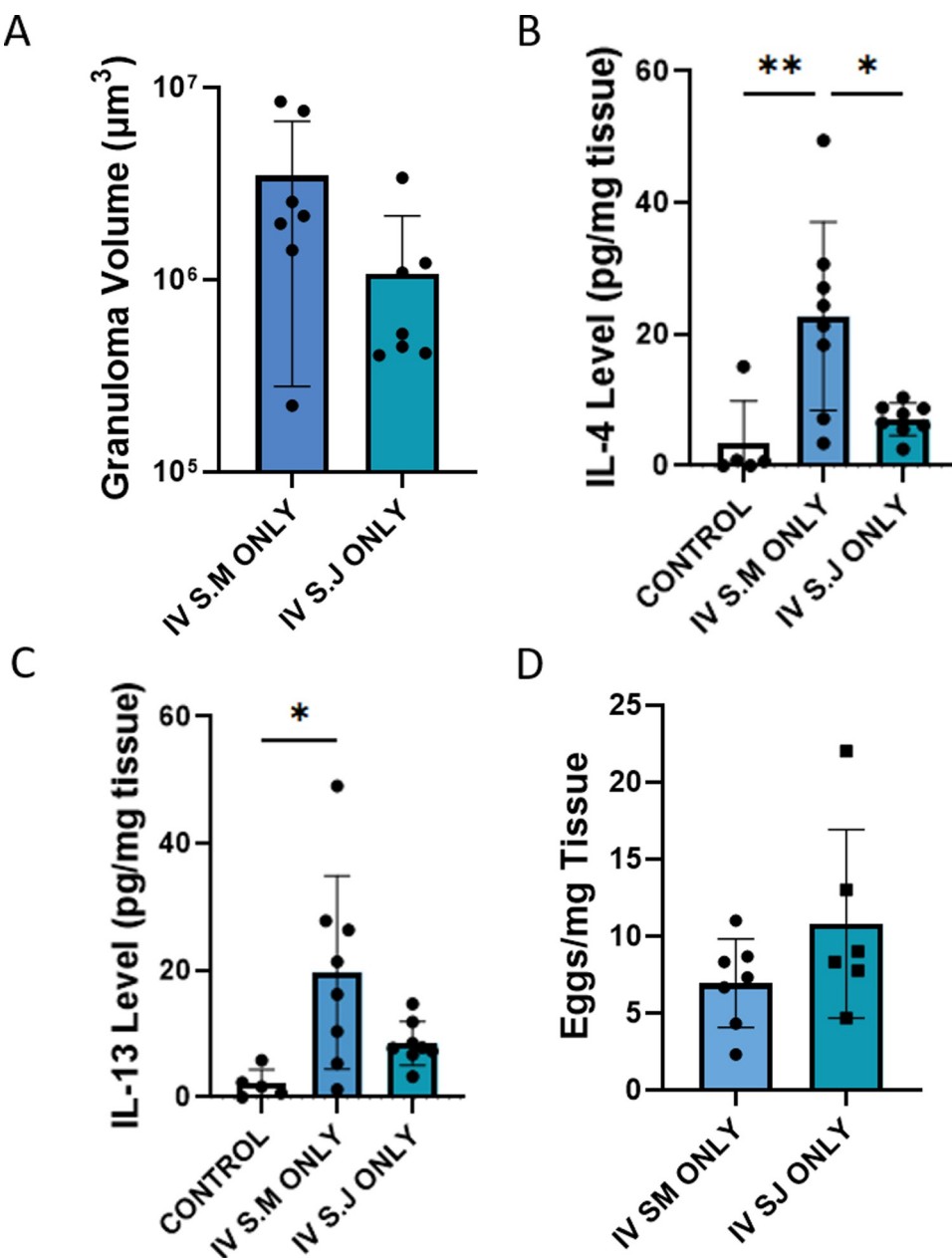

**Fig 6. Intravenous egg exposure alone from *Schistosoma* species do not induce Type-2 inflammation.** The control group data presented in panels B and C in this figure are the same control data presented in Fig 1. (A) Estimated granuloma volumes of intravenous egg alone and control mice; n = 7 per group; t-test P = NS. (B) IL-4 protein concentration in intravenous egg alone and control mice; n = 5–8 per group; ANOVA P = 0.0002; post-hoc Tukey test results shown. (C) IL-13 protein concentration in intravenous egg alone and control mice; n = 5–8 per group; ANOVA P = 0.0009; post-hoc Tukey test results shown. (D) Residual *Schistosoma* egg burden in intravenous egg challenged mice; t-test P = NS. Graphs show mean ± SD. ** P<0.01; *** P<0.005.

The vascular remodeling in *Schistosoma*-PH is more pronounced near eggs in the tissue [22], suggesting that a uniform distribution of eggs could result in more widespread vascular remodeling. However, we did not see a shift in the distribution of vessel thicknesses across the entire lung, which would be anticipated to show greater heterogeneity in the *S. japonicum* mice if there was less uniform remodeling.

Alternatively, there are likely biological differences which underlie the difference in phenotype. At the time of intravenous challenge, the eggs are alive, as would be present in humans with live worm pairs, and likely remain alive for many days within the host. It has been reported that killed eggs induce a weaker immune response [23], suggesting that products actively being produced by eggs positioned in the lungs contribute to the localized pathology, and differences in the secretions and how the host immune system reacts to these helminth products are likely to underlie the difference in phenotype observed. Teleologically, eggs laid in the portal venous system produce proteins to enable them to erode themselves through the intestinal wall, in order to reach the lumen and be excreted to the environment and propagate the parasite lifecycle, although the host immune system is also critical for egg transit [24,25]. Between the two species, there may be critical differences in the secreted products that drive the observed differences in immunity and PH. However, these pathobiologic differences are probably not driven by immunogenicity alone of the specific egg-derived products, on the basis of the observed antigenic substitutability between the two species.

Remarkably, we observed substantial antigen homology between *S. mansoni* and *japonicum*, in that either species could be used for sensitization, and the PH and inflammation severity correlated primarily with the egg species administered by intravenous challenge. Despite differences in egg biology, the substitutability of eggs of the two species at the sensitization stage prior to IV challenge suggests that the sensitizing antigens have shared homology. Based on genetic studies, it is estimated that *S. mansoni* and *S. japonicum* diverged from a common ancestor approximately 14 million years ago [26], and the two species have 67% synteny [27] (order of genetic loci on the same chromosome). As a specific example, omega-1 is a glycoprotein ribonuclease secreted by *S. mansoni* eggs, which alone can induce Th2-inflammatory response and granuloma formation [28–30]. In *S. japonicum*, the orthologue is thought to be CP1412, a glycoprotein which shares 30% homology with omega-1 and similarly causes Th2-inflammation [31], but is otherwise less well characterized. Antigenic similarities and differences of specific proteins between the two species have not been systematically investigated to our knowledge, particularly in specific pathologic contexts.

Limitations of our study include the following. Although we demonstrated that both *S. mansoni* and *S. japonicum* eggs cause Th2-inflammation, it remains unknown whether non-inflammatory mechanisms may contribute to the observed variation in results between the different endpoints. For example, although sensitization and challenge with *S. mansoni* eggs produced higher RVSPs and larger granuloma volumes than *S. japonium*, there was no significant difference in IL-4 or IL-13 level, or vascular remodeling between the two groups. This discrepancy may be due to relatively small sample sizes or experimental variability, but it may also reflect differences in immunology between species, such as the role of IL-4/IL-13 in granuloma formation, immune cell recruitment, and vascular remodeling versus vasoconstriction. In fact, granulomas around *S. japonicum* eggs are known to be more neutrophil-predominant, as opposed to those around *S. mansoni* eggs which predominantly contain macrophages and eosinophils [20,32]. It is also conceivable that IL-4/IL-13 production in the perivascular adventitial space—likely more relevant to vascular remodeling—may be masked by whole-lung assessments such as IL-4/IL-13 levels in whole lung lysates, and analysis of peri-egg granulomas.

IL-4 and IL-13 could have different functions in *Schistosoma*-induced PH which are species specific. We previously observed that mice which are deficient for both IL-4 and IL-13 were protected from *S. mansoni*-induced PH, but we found that deletion of either cytokine alone was not sufficient to induce protection indicating some role for each cytokine in the PH following this species [7]. IL-13 is implicated as a critical cytokine in the pro-fibrotic immune response to *Schistosoma* eggs [33]. In contrast, IL-4 is a major driver of the Type 2 immune

reaction, as a classical activator of Th2 CD4 T cells [34]. There could also be species-specific differences in non-immune cell responses: for example, pulmonary endothelial cells directly contact embolized eggs, and *Schistosome*-exposed endothelial cells promote leukocyte adhesion [35]. The host response may have both species and organ specificity. For example, the concept that *S. mansoni* induces more severe pulmonary disease than *S. japonicum* is in remarkable contrast to *Schistosoma* liver pathology, where *S. japonicum* is described to cause more severe liver disease than *S. mansoni* [36].

The aggregate of our findings leads to additional new insights about the pulmonary pathobiology induced by *Schistosoma* infection. The observation that *S. mansoni* produced more pronounced PH than *S. japonicum* in mice is concordant with the reported rarity of *S. japonium*-induced PAH in humans, although the extremely low incidence of reported *S. japonicu*m-induced PAH may still be out of proportion to our murine data, raising the possibility of limited screening or reporting of at-risk populations. Future studies focused on interactions between *Schistosoma* eggs and the pulmonary vasculature may elucidate mechanisms which underlie inter-species differences in PAH development.

## Supporting information

**S1 Data. Raw data for all graphs in the manuscript This Excel document contains the raw data for all graphs in the manuscript.** Each of the 14 sheets is titled with the appropriate Figure and Panel. Please refer to respective text and figure legends for details regarding each experiment.
(XLSX)

## Acknowledgments

Schistosome-infected mice were provided by the NIAID Schistosomiasis Resource Center at the Biomedical Research Institute (Rockville, MD) through NIH-NIAID Contract HHSN272201700014I for distribution through BEI Resources.

## Author Contributions

**Conceptualization:** Biruk Kassa, Rahul Kumar, Brian B. Graham.

**Data curation:** Biruk Kassa, Michael H. Lee.

**Formal analysis:** Rahul Kumar, Claudia Mickael, Rubin M. Tuder, Brian B. Graham.

**Funding acquisition:** Brian B. Graham.

**Investigation:** Biruk Kassa, Michael H. Lee, Claudia Mickael, Linda Sanders.

**Methodology:** Michael H. Lee, Rahul Kumar, Claudia Mickael, Margaret Mentink-Kane, Brian B. Graham.

**Project administration:** Brian B. Graham.

**Resources:** Margaret Mentink-Kane, Brian B. Graham.

**Software:** Brian B. Graham.

**Supervision:** Rubin M. Tuder, Brian B. Graham.

**Writing – original draft:** Biruk Kassa, Michael H. Lee.

**Writing – review & editing:** Biruk Kassa, Michael H. Lee, Rahul Kumar, Linda Sanders, Rubin M. Tuder, Margaret Mentink-Kane, Brian B. Graham.

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
