## [Decision Letter · Decision Letter 0]

17 Feb 2022

Dear Dr. Graham,

Thank you very much for submitting your manuscript "Experimental Schistosoma japonicum-induced pulmonary hypertension" for consideration at PLOS Neglected Tropical Diseases. As with all papers reviewed by the journal, your manuscript was reviewed by members of the editorial board and by several independent reviewers. The reviewers appreciated the attention to an important topic. Based on the reviews, we are likely to accept this manuscript for publication, providing that you modify the manuscript according to the review recommendations. 

Thank you for submitting this interesting and well-written article. We look forward to receiving your revisions as suggested by the reviewers.

Sincerely,

Jennifer A. Downs, M.D., Ph.D.

Associate Editor

Michael Hsieh

Deputy Editor

Thank you for submitting this interesting and well-written article. We look forward to receiving your revisions as suggested by the reviewers.

Reviewer's Responses to Questions

**Key Review Criteria Required for Acceptance?**

**Methods**

-Are the objectives of the study clearly articulated with a clear testable hypothesis stated?

-Is the study design appropriate to address the stated objectives?

-Is the population clearly described and appropriate for the hypothesis being tested?

-Is the sample size sufficient to ensure adequate power to address the hypothesis being tested?

-Were correct statistical analysis used to support conclusions?

-Are there concerns about ethical or regulatory requirements being met?

Reviewer #1: The authors presented a very interesting study comparing the potential of two different species of Schistosoma in inducing pulmonar hypertension in mice. The study is of particular relevance since provide potential explanations to the difference in epidemiology of pulmonary hypertension associated to each one of the species (japonicum and mansoni).

The methodology is adequate, considering the knowledge about SchPH models so far

Reviewer #2: The objectives are clearly articulated and the hypotheses were tested accordingly.

**Results**

-Does the analysis presented match the analysis plan?

-Are the results clearly and completely presented?

-Are the figures (Tables, Images) of sufficient quality for clarity?

Reviewer #1: The results are in accordance with the proposed methodology. Maybe the legends of the figures would benefit from a review to let the figure self-explanatory (some of the acronyms used in the figures are missing in the legend)

Reviewer #2: The figures are well presented.

The analysis follows the plan stated.

However, the description of methods (with use of references) within the RESULTS section needs to be reviewed.

**Conclusions**

-Are the conclusions supported by the data presented?

-Are the limitations of analysis clearly described?

-Do the authors discuss how these data can be helpful to advance our understanding of the topic under study?

-Is public health relevance addressed?

Reviewer #1: Conclusions are well based by the results and bring up the perspective of the findings

Reviewer #2: Yes the conclusions arise from the results.

The limitations are described correctly.

**Editorial and Data Presentation Modifications?**

Reviewer #1: Some points would benefit of clarification / exploration if the data are available.

- the authors explored the quantity of eggs per tissue and also the thickness of the granuloma peri-egg. I wonder if the authors have quantified the number of granulomas per slide - as another form to support that S. japonicum might have a different immunogenicity

- in sensitized mice, there was no difference in IL-4. Although the authors raised the potential limitation caused by the number of animals in each group, another potential reason could be a more limited role of IL4 it self in the inflammatory cascade triggered by schistosoma. The discussion would benefit of a paragraph trying to better explain the role of each one of the ILs (4 and 13) and the plausibility of the different roles according to the different specie 

- The difference in sensitized and non sensitized was an elegant way to reinforce the concept that this model is immunologically driven. It would be very nice if the authors have any data showing translation of these immunological phenomena to endothelial function. This could support potential differences between patients continuously exposed to schistosoma as compared to those already outside endemic regions

minor comment - in the author summary, japonicum is mentioned twice in the same sentence, where mansoni should be the comparator.

Reviewer #2: Remove text for methods and remove discussion from the results section.

**Summary and General Comments**

Reviewer #1: See above

Reviewer #2: This is a relevant topic and the authors should be congratulated for undertaking the study.

PLOS authors have the option to publish the peer review history of their article (what does this mean?). If published, this will include your full peer review and any attached files.

Reviewer #1: No

Reviewer #2: No

Figure Files:

Data Requirements:

Reproducibility:

References

---

## [Editor Report · Decision Letter 1]

19 Mar 2022

Dear Dr. Graham,

We are pleased to inform you that your manuscript 'Experimental Schistosoma japonicum-induced pulmonary hypertension' has been provisionally accepted for publication in PLOS Neglected Tropical Diseases.

Best regards,

Jennifer A. Downs, M.D., Ph.D.

Associate Editor

Michael Hsieh

Deputy Editor

Thank you for your thorough revisions and responsiveness to the reviewers. This is an excellent paper that will make a significant contribution to the existing literature.

---

## [Editor Report · Acceptance letter]

9 Apr 2022

Dear Dr. Graham,

We are delighted to inform you that your manuscript, "Experimental *Schistosoma japonicum*-induced pulmonary hypertension," has been formally accepted for publication in PLOS Neglected Tropical Diseases.

Best regards,

Shaden Kamhawi

co-Editor-in-Chief

Paul Brindley

co-Editor-in-Chief
